# ADVANCING AUTONOMOUS VLM AGENTS VIA VARIATIONAL SUBGOAL-CONDITIONED REINFORCEMENT LEARNING

## ABSTRACT

State-of-the-art (SOTA) reinforcement learning (RL) methods have enabled vision-language model (VLM) agents to learn from interaction with online environments without human supervision. However, these methods often struggle with learning inefficiencies when applied to complex, real-world decision-making tasks with sparse rewards and long-horizon dependencies. We propose a novel framework, Variational Subgoal-Conditioned Reinforcement Learning (VSC-RL), advancing the VLM agents in resolving challenging decision-making tasks. Fundamentally distinct from existing methods, VSC-RL reformulates the decision-making problem as a variational subgoal-conditioned RL problem with the newly derived optimization objective, Subgoal Evidence Lower BOund (SGC-ELBO), which comprises two key components: (a) maximizing the subgoal-conditioned return, and (b) minimizing the divergence from a reference goal-conditioned policy. We theoretically and empirically demonstrate that the VSC-RL can efficiently improve the learning efficiency without compromising performance guarantees. Across a diverse set of challenging benchmarks, including mobile device and web control tasks, VSC-RL consistently outperforms existing SOTA methods, achieving superior learning efficiency and performance.

## 1 INTRODUCTION

Recently, the large language models (LLMs) and vision-language models (VLMs) have demonstrated remarkable capabilities in content understanding and commonsense reasoning Yang et al. (2023a); Shen et al. (2024); Hong et al. (2023), achieving notable success in various real-world applications, such as visual question answering and visual captioning Bai et al. (2023); Chen et al. (2024). These advancements highlight the strong potential of VLMs to tackle complex real-world decision-making problems (e.g., mobile device Toyama et al. (2021) and web control Zhou et al. (2023) tasks) via building intelligent VLM agents through the advanced VLMs Zhang et al. (2023); Zheng et al. (2024). Meanwhile, after achieving impressive results in board games Silver et al. (2017) and video games Berner et al. (2019), reinforcement learning (RL) methods have been applied in online training VLM agents for tackling sequential decision-making tasks Bai et al. (2024); Qi et al. (2024).

Overall, based on the specific training paradigm, VLM agents can be categorized into three main types: prompting-based, imitation-based, and RL-based agents. Directly leveraging VLMs (e.g., Gemini-1.5-Pro Team et al. (2024) and GPT-4V OpenAI (2023)) to capture the critical information from the multimodal content, prompting-based agents aim to generate action via prompting engineering and retrieving techniques Zhang et al. (2023); Yang et al. (2023b). The performance of the prompting-based agents is usually limited, as the weights of these VLMs cannot be updated. To address this limitation, some studies Zhang & Zhang (2023); Hong et al. (2024) employ imitation learning techniques to fine-tune the open-source VLMs using human demonstrations. However, the performance of imitation-based agents is highly dependent on the quality and diversity of the demonstrations. Consequently, imitation-based agents may struggle with generalization and often underperform on out-of-distribution and unseen tasks. Recently, RL-based agents have emerged as a promising solution. By incorporating RL techniques, these agents enable VLMs to tackle complex sequential decision-making problems Bai et al. (2024); Qi et al. (2024). Nevertheless, existing RL-based agents often suffer from the learning efficiency issue in addressing challenging control

tasks with sparse reward signals and complicated goals. In many real-world scenarios, tasks require executing long sequences of actions, with rewards only provided upon successful completion. This delayed feedback poses a significant challenge for learning, fundamentally impacting the efficiency of RL-based agents. Some existing works attempt to address this issue by introducing implicit curriculum Andrychowicz et al. (2017) or hand-crafted subgoals Dayan & Hinton (1992); Chane-Sane et al. (2021). However, these existing approaches often fail to learn a proper policy in the real-world complex sequential decision-making task due to the complicated subgoal generation and curriculum design methodologies.

To address the fundamental limitations of RL-based agents mentioned above, we introduce Variational Subgoal-Conditioned RL (VSC-RL), a novel RL-based VLM agent method for enhancing learning efficiency in real-world complex sequential decision-making tasks. Based on the perspective of variational inference, VSC-RL reformulates the decision-making task as the variational subgoal-conditioned RL problem, which is later efficiently solved by utilising extensive optimization techniques. Additionally, VSC-RL utilises the significant reasoning and planning capabilities of VLM to autonomously decompose the complex goal into feasible subgoals. Given the generated subgoals, VSC-RL optimizes the objective of SubGoal-Conditioned Evidence Lower BOund (SGC-ELBO), thus effectively improving learning efficiency, consisting of (a) maximizing the subgoal-conditioned return of the target agent and (b) minimizing the subgoal-conditioned difference with the reference agent. We theoretically derive the new objective of SGC-ELBO from the original optimization objective, ensuring both improved efficiency and performance guarantees. Empirical results on various benchmarks validate our statement that VSC-RL significantly outperforms SOTA VLM agents in both sample efficiency and final performance.

In this paper, literature related to VLM agents and RL methods is discussed in Section 2. We introduce notations related to goal-conditioned RL, variational RL and subgoal generator in Section 3. In Section 4, we illustrate how to formulate the sequential decision-making problem as a variational subgoal-conditioned RL problem and derive the new optimization objective: SGC-ELBO, followed by the practical implementation of VSC-RL. In Section 5, the experimental results over various benchmarks exhibit that our VSC-RL agent can achieve superior performance compared to existing SOTAs. Overall, the main contributions of this paper are summarised as follows:

- We propose VSC-RL, a novel variational subgoal-conditioned RL method for enhancing VLM agents in resolving real-world sequential decision-making problems.
- We theoretically show that SGC-ELBO, the optimization objective of the VSC-RL, can effectively improve learning efficiency while maintaining the performance guarantee.
- We experimentally show that VSC-RL significantly outperforms various SOTAs in both learning efficiency and final performance on various challenging benchmarks.

## 2 RELATED WORKS

### 2.1 VLM AGENTS FOR DECISION-MAKING

In real-world complex control tasks requiring capacities in reasoning, planning and content understanding, it is necessary to enable agents with the vision-language models (VLMs). In particular, the VLMs can process and abstract the image and language content for challenging decision-making tasks, especially in mobile device and web control tasks Toyama et al. (2021); Liu et al. (2024). Existing VLM agents can be categorized as prompting-based, imitation-based, and RL-based agents based on the corresponding learning paradigms. Additionally, some recent works explore using VLM to enhance agents' abilities.

**Prompting-based Agent.** Leveraging the inherent reasoning and planning abilities of prosperity VLMs (e.g., Gemini-1.5-Pro Team et al. (2024) and GPT-4V OpenAI (2023)), the prompting-based agent makes decision via prompting engineering and retrieving techniques. For instance, AppAgent Zhang et al. (2023) first introduces a unified prompting-based agent method to enable the vision-language model to directly interact with mobile applications by providing the prompts with details of actions. Set-of-Marks Yang et al. (2023b) proposes a new prompting method to enhance the visual grounding ability of VLM. However, the performance of these prompting-based agents is always sensitive to the prompts required to be manually and carefully designed. Therefore, it is

challenging for the prompting-based agent to directly output the correct and desired actions to address real-world complex control problems.

**Imitation-based Agent.** The imitation-based agent learns to mimic the expert behaviours by fine-tuning the policy on human demonstration. Recently, Android in the Wild (AitW) Rawles et al. (2024) collected large-scale datasets of mobile device control tasks, enabling agents to directly learn from human experience. AutoUI Zhang & Zhang (2023) and CogAgent Hong et al. (2024) fine-tune the VLM-based policies with the AitW dataset, remarkably outperforming the prompting-based agent. In order to adapt the fine-tuned agent to the online environment, Filtered BC Pan et al. (2024b) introduces online imitation mechanisms to learn from successful online experiences. Unfortunately, these methods rely heavily on high-quality human demonstrations and often struggle to generalize to unseen tasks, limiting their application in diverse real-world scenarios.

**RL-based Agent.** Different to prompting-based and imitation-based agents, the RL-based agent can autonomously optimize the policy through trial-and-error interactions with environments, without human supervision. DigiRL Bai et al. (2024) introduces a unified offline-to-online RL framework that enables agents to learn directly from real-time interactions in dynamic environments, improving performance without the need for curated datasets. DistRL Wang et al. (2024) builds an asynchronous distributed RL system, allowing training multiple agents in parallel across different environments, thus significantly enhancing scalability and convergence speed. WebRL Qi et al. (2024) introduces a self-evolving online curriculum RL framework, enabling effective training of web agents through adaptive task generation in web control tasks. However, these RL-based agents still fundamentally suffer from the learning efficiency issue in challenging sequential decision-making tasks with sparse rewards and long horizons.

**Enhancing RL with VLM.** Recent works have shown that VLM can enhance the RL method via its remarkable capacities of reasoning, planning, and content understanding. Recent works suggest adopting VLM in reward-shaping for RL. For instance, VLM-RMs Rocamonde et al. (2023) demonstrate that VLMs can serve as effective reward models for learning complex skills. VLM can also generate the subgoals to guide the learning process for autonomous driving Pan et al. (2024a) and robot Yang et al. (2024) tasks. Nonetheless, it is still an open problem how to effectively integrate the VLM-generated subgoals into RL.

To mitigate the above issues, we present VSC-RL, which can autonomously decompose the goal into feasible subgoals by advanced VLM, and then efficiently resolve each subgoal from the principle of variational inference.

## 2.2 Goal-conditioned and Variational RL

**Goal-conditioned RL.** Sequential decision-making tasks can be viewed as the goal-conditioned RL problem Liu et al. (2022a). Based on the current state, the agent aims to find the optimal policy that guides progress toward the given goal for maximizing the return. Hindsight experience replay Andrychowicz et al. (2017) introduces an implicit curriculum learning method to enhance learning efficiency and robustness. With the perspective of divide-and-conquer, some approaches suggest guiding the agent with subgoals as intermediate reward signals via imagination Chane-Sane et al. (2021); Nair & Finn (2019) and tree-search Jurgenson et al. (2020); Parascandolo et al. (2020).

**Variational RL.** The RL problem can be viewed as the variational inference problem Levine (2018), which can be resolved by utilising extensive optimization tools, thus effectively improving the learning efficiency. Applying the expectation-maximization algorithm in the actor-critic method in RL, VIP Neumann (2011) presents a unified variational inference framework. MPO Abdolmaleki et al. (2018a;b) proposes a series of off-policy RL with entropy regulation in the manner of expectation-maximization. VDPO Wu et al. (2024) and CVPO Liu et al. (2022b) apply the variational inference techniques in addressing the RL problem with delayed signals and safety constraints, respectively.

This paper aims to show how to formulate the control problem as a variational subgoal-conditioned RL problem from the perspective of variational inference, which allows us to resolve the complicated control task by utilising extensive optimization tools.

## 3 PRELIMINARIES

**Finite-Horizon Goal-Conditioned MDP.** We formulate the RL problem as the finite horizon goal-conditioned Markov Decision Process (MDP), denoted by the tuple $< \mathcal{G}, \mathcal{S}, \mathcal{A}, \mathcal{R}, \mathcal{T}, H >$ where $\mathcal{G}$ is the goal set, $\mathcal{S}$ is the state space, $\mathcal{A}$ is the action space, $\mathcal{T} : \mathcal{S} \times \mathcal{A} \times \mathcal{S} \rightarrow [0, 1]$ is the dynamic function, $\mathcal{R}$ is the reward function and $H$ is the horizon. At each timestep $t$, the agent takes action $a_t \in \mathcal{A}$ (e.g., typing text, press button or slide the screen) based on its policy $\pi : \mathcal{S} \times \mathcal{G} \times \mathcal{A} \rightarrow [0, 1]$, the current screenshot $s_t \in \mathcal{S}$, and a specific goal $g \in \mathcal{G}$ (e.g., search a new TV at Best Buy) selected in the beginning of each episode. The agent only receives the reward $r_t = 1$ if the goal $g$ is accomplished, otherwise the reward $r_t = 0$. The objective of the agent is to find the policy $\pi$ which can accomplish all goals from the goal set $\mathcal{G}$ within the finite horizon $H$.

**Variational RL.** RL can be viewed as a variational inference problem. We denote the optimality of a trajectory $\tau$ is the event $O$, and the corresponding probability of the trajectory optimality is denoted as $p(O|\tau) \propto \exp\left(\frac{\mathcal{J}(\tau)}{\alpha}\right)$ where $J(\tau) := \sum_{t=0}^{H} \gamma^t \mathcal{R}(s_t, a_t)$ is the discounted return of the trajectory $\tau := \{s_t, a_t\}_{t=0}^{H}$ and $\alpha$ is the temperature. Therefore, the objective transforms to finding a policy $\pi$ with the highest log evidence: $\max_\pi \log p_\pi(O)$. Furthermore, the Evidence Lower BOund of the objective is:

$$\mathbb{E}_{\tau \sim q(\tau)} [\log p(O|\tau)] - \mathrm{KL}(q(\tau)||p_\pi(\tau)), \tag{1}$$

where $p_\pi(\tau)$ is the trajectory distribution induced by policy $\pi$, $q(\tau)$ is the prior trajectory distribution and KL is the Kullback-Leibler divergence. Thus, the objective of Variational RL is maximizing the ELBO (Equation (1)).

**Subgoal Generator.** For challenging control tasks with sparse and long-term reward signals, it is difficult to learn a useful policy that arrives at the final goal within a finite horizon. Therefore, subgoal generation is particularly useful in providing the intermediate signals to facilitate learning. Then, we introduce the assumption of the existence of subgoals for the given goal, aiming to bring the goal-conditioned RL problem to the subgoal-conditioned RL problem as follows.

**Assumption 3.1** (Existence of Subgoals). *Given a trajectory $\tau$ and the corresponding goal $g$, it always exists a sequence of sub-trajectories and corresponding subgoals $\{\tau_i, sg_i\}_{i=1}^{N}(1 \leq N \leq H)$ induced from the $\tau$ and $g$.*

Commonly adopted in literature Sutton et al. (1999), the above assumption is mild and usually holds. For instance, when $N = 1$, the subgoals and sub-trajectories are the original goal and trajectory, respectively. When $N = H$, each sub-trajectory is composed of one single transition-tuple $(s_t, a_t, r_t, s_{t+1})$ with its corresponding subgoal.

## 4 OUR APPROACH: VSC-RL

In this section, we present our approach, Variational Subgoal-Conditioned Reinforcement Learning (VSC-RL) for enhancing VLM agents in solving real-world decision-making tasks. First, we formulate the sequential decision-making task as the variational goal-conditioned RL problem (Section 4.1). Next, we derive the new subgoal-conditioned optimization objective, SGC-ELBO, consisting of (a) maximizing the subgoal-conditioned return (Proposition 4.1) and (b) minimizing the subgoal-conditioned difference (Proposition 4.2). We also theoretically show the derivation of new optimization objective, ensuring both improved learning efficiency and performance guarantees. In Section 4.3, we demonstrate that VLMs can effectively generate feasible subgoals from the complex goal for VSC-RL. The practical implementation is illustrated in Section 4.4. We present the overall pipeline of VSC-RL in Figure 1, and the pseudo-code of VSC-RL is summarised in Algorithm 1.

### 4.1 PROBLEM FORMULATION

We first formulate the sequential decision-making as the variational goal-conditioned RL problem. In this context, similar to Equation (1), the objective is to find a goal-conditioned policy $\pi$ with the highest log evidence: $\max_\pi \log p_\pi(O|g)$ for a given goal $g$. Then, we have the Goal-Conditioned ELBO (GC-ELBO) of $\log p_\pi(O|\tau, g)$ as follows:

$$\text{GC-ELBO}(\pi, \pi_{\text{ref}}, g) = \mathop{\mathbb{E}}_{\tau \sim p_\pi(\tau|g)} \left[\log p(O|\tau, g)\right] - \text{KL}(p_\pi(\tau|g)||p_{\pi_{\text{ref}}}(\tau|g)), \qquad (2)$$

where $p_{\pi_{\text{ref}}}(\tau|g)$ is the prior trajectory distribution of the goal-conditioned reference policy $\pi_{\text{ref}}$ for the given goal $g$. Therefore, from Equation (2), the objective becomes maximizing the GC-ELBO: $\max_\pi \text{GC-ELBO}(\pi, \pi_{\text{ref}}, g)$.

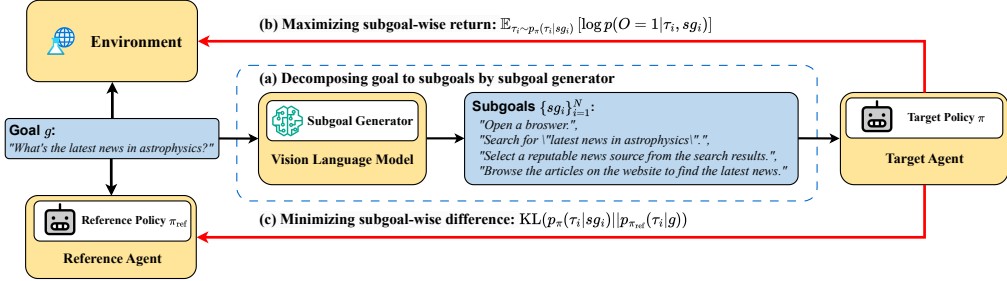

Figure 1: The pipeline of VSC-RL. (a) VLM autonomously decomposes the goal $g$ to the subgoals $\{sg_i\}_{i=1}^N$. VSC-RL optimizes the objective of SGC-ELBO consisting of (b) maximizing the subgoal-conditioned return and (c) minimizing the subgoal-conditioned difference.

---

**Algorithm 1** VSC-RL

---

**Input:** goal $g$, subgoal generator VLM, reference policy $\pi_{\text{ref}}$, target policy $\pi$, value function $V$;
**for** Epoch $= 1, \cdots$ **do**
    Generate Subgoals $\{sg_i\}_{i=1}^N \sim \text{VLM}(g)$
    Collect $(\tau_i, sg_i)_{i=1}^N$ from $\pi$ for the given goal $g$
    # Optimize the SGC-ELBO (Equation (4))
    Maximize Subgoal-conditioned Return via Equation (5) and Equation (6)
    Minimize Subgoal-conditioned Behavior Difference via Equation (7)
**end for**
**Output:** updated policy $\pi$

---

## 4.2 VARIATIONAL SUBGOAL-CONDITIONED RL

With the assumption of the subgoals (Assumption 3.1), we demonstrate that the former term of GC-ELBO (Equation (2)) is equivalent to the maximizing subgoal-conditioned RL objective (Proposition 4.1) and the latter term of GC-ELBO can be transformed to the minimizing subgoal-conditioned difference (Proposition 4.2).

Based on Equation (2), we show that the former term, $\mathbb{E}_{\tau \sim p_\pi(\tau|g)}\left[\log p(O|\tau, g)\right]$, can be reformulated in the subgoal-conditioned RL objective with shorter-horizon in the following Proposition 4.1.

**Proposition 4.1** (Subgoal-Conditioned Optimization Objective, Proof in Proposition C.1)**.** *Given a goal $g$ with corresponding subgoals $\{sg_i\}_{i=1}^N$ and a subgoal-conditioned target policy $\pi$, the objective of*

$$\max_\pi \mathop{\mathbb{E}}_{\tau \sim p_\pi(\tau|g)} \left[\log p(O|\tau, g)\right]$$

*is equivalent to the objective of*

$$\max_\pi \sum_{i=1}^N \left[\mathop{\mathbb{E}}_{\tau_i \sim p_\pi(\tau_i|sg_i)} \left[\log p(O|\tau_i, sg_i)\right]\right].$$

In the above proposition, the goal-wise objective has been transformed into the subgoal-conditioned objective, which is composed of $N$ subgoals with corresponding shorter horizons. Thus, the agent can learn from these reward signals from the subgoals, thus effectively improving the learning efficiency Jiang & Agarwal (2018).

Next, we show that the latter term in Equation (2), $\text{KL}(p_\pi(\tau|g)||p_{\pi_{\text{ref}}}(\tau|g))$, has the subgoal-conditioned upper bound in the following proposition.

**Proposition 4.2** (Subgoal-conditioned Difference Bound, Proof in Proposition C.2). *Given goal-conditioned reference policy $\pi_{ref}$ and subgoal-conditioned target policy $\pi$, the goal-conditioned KL divergence of a given goal $g$ has the upper bound of subgoal-conditioned KL divergence of corresponding subgoals $\{sg_i\}_{i=1}^N$ as follows:*

$$KL(p_\pi(\tau|g)||p_{\pi_{ref}}(\tau|g)) \leq \sum_{i=1}^N \left[ KL(p_\pi(\tau_i|sg_i)||p_{\pi_{ref}}(\tau_i|g)) \right].$$

Therefore, from Proposition 4.2, we can directly minimize the $N$ subgoal-conditioned KL divergences, which is the upper bound of the goal-conditioned KL divergence.

Based on Proposition 4.1 and Proposition 4.2, the newly-derived optimization objective of SubGoal-Conditioned ELBO (SGC-ELBO) is as follows:

$$\text{SGC-ELBO}(\pi, \pi_{\text{ref}}, sg_i, g) = \mathbb{E}_{\tau_i \sim p_\pi(\tau_i|sg_i)} \left[ \log p(O|\tau_i, sg_i) \right] - \text{KL}(p_\pi(\tau_i|sg_i)||p_{\pi_{\text{ref}}}(\tau_i|g)). \quad (3)$$

Equation (3) consists of two key components: (a) maximizing the subgoal-conditioned return of the target policy $\pi$ and (b) minimizing the subgoal-conditioned difference between $\pi$ and the reference policy $\pi_{\text{ref}}$. Therefore, the agent can directly learn to resolve the subgoal $sg_i$ with a shorter horizon requirement, effectively improving the learning efficiency. The newly derived optimization objective of SGC-ELBO (Equation (3)) can improve learning efficiency without compromising performance guarantees.

### 4.3 AUTONOMOUS SUBGOAL GENERATION VIA VISION-LANGUAGE MODELS

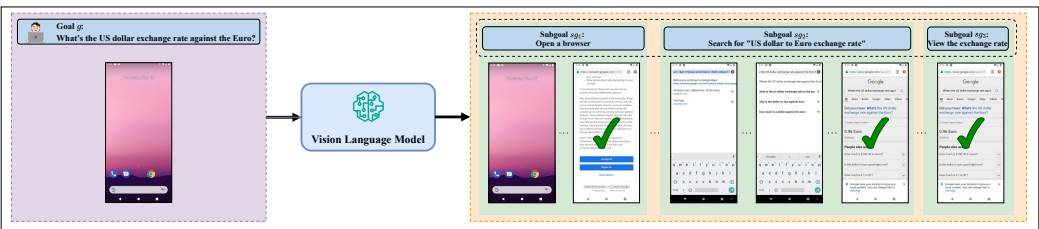

Figure 2: Autonomous subgoal generation in AitW task. The VLM autonomously decomposes the goal of the complicated mobile device control task into easily achievable subgoals.

For real-world complex decision-making, it is challenging to handcraft and design the subgoals for each goal manually. VLM has exhibited a unique reasoning ability in image captioning, visual question answering, and multimodal reasoning via integrating and interpreting visual and textual information to derive meaningful insights for VLM agents. Therefore, we use VLM as the subgoal generator, which autonomously decomposes the given goal $g$ into the feasible subgoals $\{sg_i\}_{i=1}^N$. As demonstrated in the AitW task example (Figure 2), VLM can autonomously decompose the goal: *"What's the US dollar exchange rate against the Euro?"* into more specific and easily solvable subgoals, including *"Open a browser"*, *"Search for "US dollar to Euro exchange rate"*, and *"View the exchange rate"*. Therefore, we can tell that the VLM can serve as the subgoal generator in the general case. Additionally, we also provide the example of subgoal generation in Appendix J. In the context of VLM as the autonomous vision-language subgoal generator, the optimization objective (Equation (3)) can be written as

$$\max_\pi \left[ \sum_{\{sg_i\}_{i=1}^N \sim \text{VLM}(g)} \left[ \text{SGC-ELBO}(\pi, \pi_{\text{ref}}, sg_i, g) \right] \right], \quad (4)$$

where subgoals $\{sg_i\}_{i=1}^N$ are generated by a VLM through prompting with the original goal $g$.

### 4.4 PRACTICAL IMPLEMENTATION

As a unified RL-based agent framework, most existing RL-based methods can easily be embedded in VSC-RL. In this paper, we mainly consider the mobile device and web control tasks to evaluate

the VSC-RL, a representative and challenging real-world decision-making task which has drawn attention recently. Specifically, the reference agent $\pi_{\text{ref}}$ and target agent $\pi$ are both initialised as the AutoUI-Base agent Zhang & Zhang (2023), which is pre-trained on the Android in the Wild (AitW) datasets. To maximize the subgoal-conditioned RL objective in Equation (3), VSC-RL uses the Advantage-Weighted Regression (AWR) algorithm Peng et al. (2019) modified by DigiRL Bai et al. (2024) as follows:

$$\arg\max_{\pi} \mathbb{E}_{s,a,sg_i \sim \mathcal{D}} \left[ \log \pi(a|s, sg_i) \exp \left( \frac{A(s, a, sg_i)}{\beta} \right) \right], \tag{5}$$

where $\mathcal{D}$ is the replay buffer, $\beta$ is the hyperparameter and $A(s, a, sg_i) := R_i - V(s, a, sg_i)$ is the advantage function. Specifically, $V(s, a, sg_i)$ is the subgoal-conditioned value function which aims to predict the return $R_i$ of the subgoal $sg_i$ as follows:

$$\arg\min_{V} \mathbb{E}_{s,a,sg_i,R_i \sim \mathcal{D}} \left[ ||R_i - V(s, a, sg_i)|| \right], \tag{6}$$

where $R_i$ is the binary return evaluated by the VLM Pan et al. (2024b). VSC-RL minimizes the subgoal-conditioned KL divergence in Equation (3) via imitation loss as follows:

$$\arg\max_{\pi} \mathbb{E}_{\substack{a_{\text{ref}} \sim \pi_{\text{ref}}(\cdot|s,g) \\ s,sg_i,g \sim \mathcal{D}}} \left[ \log \pi(a_{\text{ref}}|s, sg_i) \right], \tag{7}$$

where $a_{\text{ref}}$ is the reference action. Similar to DigiRL Bai et al. (2024), VSC-RL additionally learns the instruction-level value function for filtering the sub-trajectories and accelerating the learning. VSC-RL adopts Gemini-1.5-Pro Team et al. (2024) as the subgoal generator. Specifically, we in-context prompt the VLM to generate the subgoals for a given goal, including human demonstration as examples. The prompt example is provided in Appendix J. Overall, the pseudo-code of VSC-RL is summarised in Algorithm 1.

## 5 EXPERIMENTS

In this section, we empirically demonstrate that our VSC-RL can achieve better sample efficiency and a higher success rate than various state-of-the-art (SOTA) agents in various challenging benchmarks, including AitW Rawles et al. (2024) and WebArena-Lite Liu et al. (2024). We also present the ablation results for evaluating the key components of VSC-RL. The implementation details and hyperparameter settings are listed in Appendix B. Additionally, we present the additional experiments on MiniGrid Chevalier-Boisvert et al. (2019) in Appendix D. Additional experiments investigating the subgoal generator in VSC-RL are presented in Appendix E.

### 5.1 EXPERIMENTAL SETTINGS

**Benchmarks.** For the complex and challenging problem, we mainly consider AitW General and Web Shopping tasks Rawles et al. (2024), two kinds of the most challenging device control tasks for evaluation. The horizons of General and Web Shopping tasks are set to 10 and 20 steps, respectively. The success of the task is autonomously evaluated by the Gemini-1.5-Pro Team et al. (2024) via the in-context prompting approach. We also evaluate VSC-RL on WebArena-Lite Liu et al. (2024), a human-verified subset of the WebArena benchmark Zhou et al. (2023) containing 165 realistic web tasks across five websites. Each task involves complex HTML-based observations with 30 steps of horizon. Following WebRL Qi et al. (2024), we adopt the pretrained outcome-supervised reward model (ORM) to autonomously evaluate the task's success.

**Baselines.** For the AitW tasks, we compare our VSC-RL with various SOTA baselines, including prompting-based agents (Set-of-Marks Yang et al. (2023b) and AppAgent Zhang et al. (2023)), imitation-based agents (AutoUI Zhang & Zhang (2023), CogAgent Hong et al. (2024) and Filtered BC Pan et al. (2024b)) and RL-based agents (DigiRL Bai et al. (2024)). Each method is tested on 3 independent runs, consistent with existing works Bai et al. (2024). For the WebArena-Lite tasks, we compare VSC-RL with several SOTA baselines adapted to web environments, including supervised fine-tuning (SFT), Filtered BC Pan et al. (2024b), and RL-based agents (AWR Peng et al. (2019), DigiRL Bai et al. (2024), and WebRL Qi et al. (2024)). Specifically, to ensure a fair comparison, we remove the curriculum learning component of WebRL so that all methods are trained solely on the same task set. Each method is tested on 1 single run, consistent with existing works Qi et al. (2024).

## 5.2 EXPERIMENTAL RESULTS AND ANALYSIS

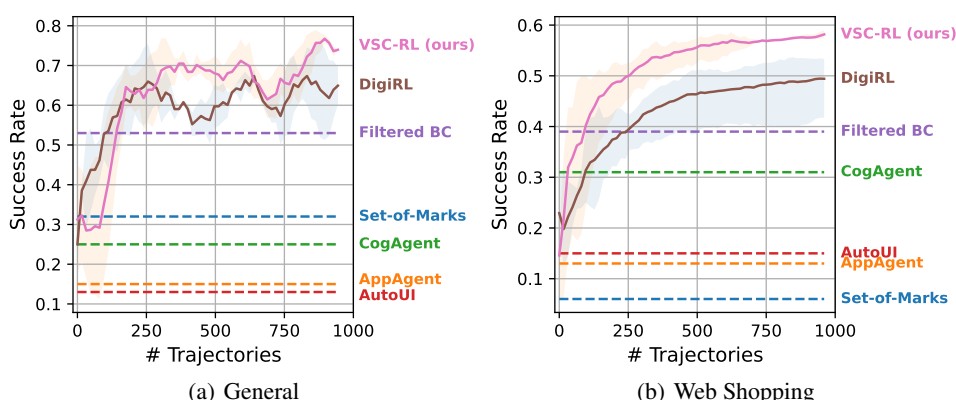

(a) General

(b) Web Shopping

Figure 3: Learning curves on AitW (a) General and (b) Web Shopping tasks.

Table 1: The performance on General and Web Shopping tasks. The best performance is **highlighted**.

| Task | | Set-of-Marks | AppAgent | CogAgent | AutoUI | Filtered BC | DigiRL | VSC-RL (ours) |
|---|---|---|---|---|---|---|---|---|
| General | Train | 32.3% | 14.6% | 25.0% | 12.5% | 53.5% | 64.9% | **73.9%** |
| | Test | 16.7% | 16.7% | 25.0% | 14.6% | 62.5% | 67.7% | **72.9%** |
| Web Shopping | Train | 6.3% | 5.2% | 31.3% | 14.6% | 53.6% | 55.3% | **64.6%** |
| | Test | 11.5% | 8.3% | 38.5% | 17.7% | 54.2% | 41.3% | **58.3%** |

**AitW.** The learning curves of AitW General and Web Shopping are summarised in Figure 3. Overall, our VSC-RL outperforms other baselines significantly in both the General and Web Shopping tasks. The RL-based agents (DigiRL and VSC-RL) both show leading performance in the AitW General task. After reaching a similar performance of $65.0\%$ success rate with DigiRL in 250 trajectories, our VSC-RL outperforms all baselines significantly, arriving at the best final performance of $75\%$ success rate. Similarly, RL-based agents (DigiRL and VSC-RL) dominate all other types of agents remarkably in the Web Shopping task. Specifically, our VSC-RL can finally achieve around $60.0\%$ success rate, significantly outperforming $50.0\%$ success rate of DigiRL. We also evaluate the generalization of our VSC-RL on the test datasets, including a range of unseen tasks, respectively. The results summarised in Table 1 tell us that our VSC-RL shows significant superiority in both the train and test datasets. Especially, in the general tasks, VSC-RL performs approximately $+13.9\%$ and $+7.7\%$ better than the second-best baseline on the train and test datasets, respectively. Similarly, VSC-RL achieves the best performance on both the train and test datasets of web shopping tasks. Overall, VSC-RL can exhibit consistent performance on unseen tasks, showing remarkable generalization ability.

**WebArena-Lite.** As shown in Figure 4, we present the learn curves on the WebArena-Lite. The imitation-based agents, SFT and Filtered BC achieve relatively limited performance, with success rates stagnating around $20.6\%$ and $23.0\%$, respectively. AWR achieves approximately $28.5\%$ success rate via solely leveraging the offline RL technique, which is still limited compared to the online RL methods. DigiRL and WebRL exhibit similar performance trends, both plateauing around a $31.0\%$ success rate. Our VSC-RL consistently outperforms the other methods, achieving the highest success rate of approximately $34.5\%$. Overall, our VSC-RL can achieve superior learning efficiency and final performance on the WebArena-Lite. Specifically, the additional results shown in Appendix F demonstrate that our VSC-RL can achieve the best performance across all types of tasks on the WebArena-Lite.

**Ablation Results of VSC-RL.** The ablation results on the Web Shopping task, shown in Table 2, evaluate the impact of various key components of VSC-RL. Introducing subgoals to AutoUI can yield a marginal improvement $(+1.6\%)$ on the average success rate from $16.2\%$ to $17.8\%$, which offers limited benefit without involving the RL process. For VSC-RL, removing subgoals entirely leads to a substantial performance drop $(-7.8\%)$ compared to providing limited (50% less) subgoals

Figure 4: Learning curves on WebArena-Lite.

Table 2: Ablation results of VSC-RL on the Web Shopping. The best performance is **highlighted**.

| Method | Web Shopping | | |
|---|---|---|---|
| | Train | Test | Average |
| AutoUI | 14.6% | 17.7% | 16.2% |
| w/ subgoals | 16.7% | 18.8% | 17.8% |
| VSC-RL | **64.6%** | **58.3%** | **61.5%** |
| w/o subgoals | 55.2% | 52.1% | 53.7% |
| w/ limited (50% less) subgoals | 57.3% | 55.2% | 56.3% |
| w/o policy gradient (Equation (5)) | 56.3% | 51.0% | 53.7% |
| w/o imitation loss (Equation (7)) | 55.2% | 46.9% | 51.1% |

$(-5.2\%)$. These results imply that subgoals can efficiently improve performance by providing immediate informative reward signals in the RL process. Additionally, we investigate the different optimization components of SGC-ELBO(Equation (3)) in VSC-RL. Removing the policy gradient (Equation (5)) and imitation loss (Equation (7)) result in decreased performance with $-7.8\%$ and $-10.4\%$, respectively. Overall, each optimization component of VSC-RL contributes meaningfully to its effectiveness, aligning with our main statements in Section 4.2. Ablation studies on subgoal quality and subgoal generator are provided in Appendix G and Appendix H, respectively. We also analyze the failure cases of VSC-RL in Appendix I.

## 5.3 LIMITATIONS AND CHALLENGES

We have empirically demonstrated that our VSC-RL can effectively address the learning efficiency issue commonly existing in complex sequential decision-making tasks. However, there are still some limitations and challenges in VSC-RL, as discussed below.

**Fine-tuning VLM as Subgoal Generator.** Benefiting from the general reasoning ability of the proprietary VLM, we empirically found that the performance of VSC-RL is improved by the feasible subgoals. However, for the control task from a specific domain, it is worth fine-tuning the open-source VLM as the subgoal generator.

**Hierarchical RL Approaches.** Additionally, the VLM in VSC-RL cannot only be viewed as the subgoal generator, but also as the high-level policy in the context of hierarchical RL. It is valuable to investigate enhancing VSC-RL with the hierarchical RL approaches Zhao et al. (2024); Zhou et al. (2024) with re-planning ability (Zeng et al., 2025).

**Future Challenging Applications.** In this work, we mainly consider the mobile device and web control tasks, two representative complex control problems, as the evaluation benchmarks. The theoretical and empirical results presented in this work imply that VSC-RL has great potential in addressing other challenging open problems, such as MCP-enabled control tasks (Liu et al., 2025; Yan et al., 2025; Lumer et al., 2025).

## 6 CONCLUSION

This work investigates advancing VLM agents in resolving real-world complex sequential decision-making tasks. Existing promising RL-based agents often suffer from the learning efficiency issue in solving tasks with complicated goals and sparse reward signals. To address this issue, we propose VSC-RL, which can autonomously decompose the goal to subgoals and resolve them efficiently. VSC-RL reformulates the decision-making task as a variational subgoal-conditioned RL problem with the new derived optimization objective of SGC-ELBO, thus effectively improving the learning efficiency without compromising the performance guarantee. In various benchmarks, especially in challenging mobile device and web control tasks, we empirically show that VSC-RL exhibits significant performance improvement and learning efficiency, remarkably outperforming existing methods.

## ETHICS STATEMENT

We affirm that all authors have read and adhere to the ICLR Code of Ethics. This work does not involve human or animal subjects, sensitive personal data, or privacy risks. The use of LLMs was limited to writing support and language refinement. LLMs were not used in the design of algorithms, the development of theoretical results, or the execution of experiments, ensuring that all core scientific contributions are entirely the work of the authors.

## REPRODUCIBILITY STATEMENT

We provide the implementation detail in Appendix B. We also provide the codebase reproducing our experimental results in the supplementary material.

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

## A  LLM USAGE STATEMENT

In this work, the use of LLMs was restricted to writing support and language refinement. Specifically, LLMs assisted in enhancing the clarity and coherence of the manuscript. LLMs were not used in the design of algorithms, the development of theoretical results, or the execution of experiments, ensuring that all core scientific contributions are entirely the work of the authors.

## B  IMPLEMENTATION DETAILS

As shown in Table 3, we summarize LLMs and VLMs used in VSC-RL. For AitW tasksRawles et al. (2024), we built our VSC-RL on the open repository of DigiRL Bai et al. (2024). Hyperparameter settings are listed in Table 4, and each run of VSC-RL takes approximately 24 hours on 1 NVIDIA GeForce RTX 4090 GPU and 8 Intel Xeon CPUs. For WebArena-Lite tasks Liu et al. (2024), we built our VSC-RL on the open repository of WebRL Qi et al. (2024) and followed the online RL loop of interaction, filtering, and update. To ensure a fair comparison, we remove WebRL's curriculum learning component so that all methods are trained solely on the same task set. For VSC-RL and other baselines, we apply the same actor perplexity-based filtering strategy as WebRL to select replay data, ensuring consistency in experience quality. Hyperparameter settings are listed in Table 5, and each run of VSC-RL takes approximately 24 hours on 8 NVIDIA GeForce RTX 4090 GPUs and 8 Intel Xeon CPUs. Specifically, we report some results on AitW (Set-of-Marks, AppAgent, CogAgent, AutoUI, and Filtered BC) and WebArena-Lite (SFT, Filtered BC, and AWR) from the literature Bai et al. (2024); Qi et al. (2024).

Table 3: The summary of LLMs and VLMs used in VSC-RL.

| Component | Task | | Description |
| --- | --- | --- | --- |
| | AitW | WebArena-Lite | |
| Subgoal Generator | Gemini-1.5-Pro | Gemini-1.5-Pro | Autonomously decompose the goal into subgoals. |
| Reference Actor $\pi_{ref}$ | AutoUI-Base | Llama-3.1-8B | Provide reference action for imitation. |
| Target Actor $\pi$ | AutoUI-Base | Llama-3.1-8B | Make decisions to maximize subgoal-conditioned return. |
| Evaluator | Gemini-1.5-Pro | Llama-3.1-8B | Autonomously evaluate the goal's or subgoal's success. |

Table 4: Hyperparameters settings of VSC-RL on AitW tasks.

| Hyperparameter | Value |
| --- | --- |
| Batch Size | 4 |
| Total Trajectories | 1,000 |
| Discount Factor | 0.5 |
| Learning Rate | 1e-4 |
| Update Epoch (Actor Equation (5)) | 20 |
| Update Epoch (Critic Equation (6)) | 5 |
| Update Epoch (Actor Equation (7)) | 20 |
| Update Epoch (Instruction-level Critic) | 5 |
| Maximum Gradient Norm | 0.01 |

Table 5: Hyperparameter settings of VSC-RL on WebArena-Lite tasks.

| Hyperparameter | Value |
| --- | --- |
| Batch Size | 128 |
| Total Trajectories | 1,000 |
| Discount Factor | 0.9 |
| Learning Rate | 1e-6 |
| Update Epoch (Actor Equation (5)) | 1 |
| Update Epoch (Critic Equation (6)) | 1 |
| Update Epoch (Actor Equation (7)) | 1 |
| Maximum Gradient Norm | 1.0 |

## C  THEORETICAL ANALYSIS

**Proposition C.1** (Subgoal-Conditioned Optimization Objective). *Given a goal g with corresponding subgoals $\{sg_i\}_{i=1}^N$ and a subgoal-conditioned target policy π, the objective of*

$$\max_{\pi} \mathbb{E}_{\tau \sim p_\pi(\tau|g)} \left[ \log p(O|\tau, g) \right]$$

*is equivalent to the objective of*

$$\max_{\pi} \sum_{i=1}^N \left[ \mathbb{E}_{\tau_i \sim p_\pi(\tau_i|sg_i)} \left[ \log p(O|\tau_i, sg_i) \right] \right].$$

*Proof.* We have

$$\log p(O|\tau, g) \propto \exp\left( \frac{\mathcal{J}(\tau|g)}{\alpha} \right) = \exp\left( \frac{\sum_{i=1}^N [\mathcal{J}(\tau_i|sg_i)]}{\alpha} \right).$$

So, we have

$$\mathbb{E}_{\tau \sim p_\pi(\tau|g)} \left[ \log p(O|\tau, g) \right]$$
$$= \mathbb{E}_{\tau \sim p_\pi(\tau, g)} \left[ \mathcal{J}(\tau, g) \right]$$
$$= \mathbb{E}_{\tau \sim \prod_{i=1}^N p_\pi(\tau_i, sg_i)} \left[ \mathcal{J}(\tau|g) \right]$$
$$= \mathbb{E}_{\tau \sim \prod_{i=1}^N p_\pi(\tau_i, sg_i)} \left[ \sum_{i=1}^N \mathcal{J}(\tau_i|sg_i) \right]$$
$$= \sum_{i=1}^N \left[ \mathbb{E}_{\tau_i \sim p_\pi(\tau_i, sg_i)} \left[ \mathcal{J}(\tau_i|sg_i) \right] \right]$$

Due to the fact that

$$\log p(O|\tau_i, sg_i) \propto \exp\left( \frac{\mathcal{J}(\tau_i|sg_i)}{\alpha} \right).$$

Therefore, we have

$$\max_{\pi} \mathbb{E}_{\tau \sim p_\pi(\tau|g)} \left[ \log p(O|\tau, g) \right] \Rightarrow \max_{\pi} \sum_{i=1}^N \left[ \mathbb{E}_{\tau_i \sim p_\pi(\tau_i|g_i)} \left[ \log p(O|\tau_i, sg_i) \right] \right]$$

$\square$

**Proposition C.2** (Subgoal-conditioned Difference Bound). *Given goal-conditioned reference policy $\pi_{ref}$ and subgoal-conditioned target policy π, the goal-conditioned KL divergence of a given goal g has the upper bound of subgoal-conditioned KL divergence of corresponding subgoals $\{sg_i\}_{i=1}^N$ as follows:*

$$KL(p_\pi(\tau|g)||p_{\pi_{ref}}(\tau|g)) \leq \sum_{i=1}^N \left[ KL(p_\pi(\tau_i|sg_i)||p_{\pi_{ref}}(\tau_i|g)) \right].$$

*Proof.* We have

$$p_\pi(\tau|g) = \rho(s_0) \prod_{t=0}^H P(s_{t+1}|s_t, a_t) \pi(a_t|s_t, g),$$
$$= \prod_{i=1}^N p_\pi(\tau_i|sg_i).$$
$$\leq p_\pi(\tau_i|sg_i)(i = 1, \cdots, N)$$

Similarly, we have

$$p_{\pi_{\text{ref}}}(\tau|g) = \prod_{i=1}^{N} p_{\pi_{\text{ref}}}(\tau_i|g)$$

Therefore,

$$
\begin{aligned}
&\text{KL}(p_\pi(\tau|g)||p_{\pi_{\text{ref}}}(\tau|g)) \\
&= \underset{\tau \sim p_\pi(\tau|g)}{\mathbb{E}} \left[\log p_\pi(\tau|g) - \log p_{\pi_{\text{ref}}}(\tau|g)\right] \\
&= \underset{\tau \sim p_\pi(\tau|g)}{\mathbb{E}} \left[\sum_{i=1}^{N} \log p_\pi(\tau_i|sg_i) - \sum_{i=1}^{N} \log p_{\pi_{\text{ref}}}(\tau_i|g)\right] \\
&= \sum_{i=1}^{N} \left[\underset{\tau \sim p_\pi(\tau|g)}{\mathbb{E}} \left[\log p_\pi(\tau_i|sg_i) - \log p_{\pi_{\text{ref}}}(\tau_i|g)\right]\right] \\
&\leq \sum_{i=1}^{N} \left[\underset{\tau_i \sim p_\pi(\tau_i|sg_i)}{\mathbb{E}} \left[\log p_\pi(\tau_i|sg_i) - \log p_{\pi_{\text{ref}}}(\tau_i|g)\right]\right] \\
&= \sum_{i=1}^{N} \left[\text{KL}(p_\pi(\tau_i|sg_i)||p_{\pi_{\text{ref}}}(\tau_i|g))\right]
\end{aligned}
$$

$\square$

# D ADDITIONAL EXPERIMENTS: MINIGRID

We also evaluate our VSC-RL on the toy vision-language decision-making tasks, MiniGrid Chevalier-Boisvert et al. (2019). We select the PPO Schulman et al. (2017) as the baseline, and we apply VSC-RL in the PPO for a fair comparison. We built our VSC-RL on the open repository of babyAI Chevalier-Boisvert et al. (2019), hyperparameter settings are listed in Table 6. Overall, as shown in Figure 5, our VSC-RL outperforms the baseline in all tasks remarkably, especially in the difficult task with the increasing number of rooms. From the result of MultiRoom-N2-v0 shown in Figure 5(a), we can tell that although PPO and VSC-RL both successfully reach $100\%$ success rate, our VSC-RL shows a better sample efficiency. For MultiRoom-N4-v0 (Figure 5(b)) and MultiRoom-N6-v0 (Figure 5(c)) where PPO is not able to learn any useful policy, while VSC-RL exhibits strong performance of $100\%$ and $80\%$ success rate, respectively.

Table 6: Hyperparameter settings of VSC-RL on MiniGrid.

| Hyperparameter | Value |
| --- | --- |
| Batch Size | 256 |
| Total Steps | 200,000 |
| Discount Factor | 0.99 |
| Learning Rate | 1e-3 |
| Network Layers (Image) | 3 |
| Network Layers (Text) | 1 |
| Network Layers (Actor) | 2 |
| Network Layers (Critic) | 2 |
| Update Epoch (Actor Equation (5)) | 4 |
| Update Epoch (Critic Equation (6)) | 4 |
| Update Epoch (Actor Equation (7)) | 4 |
| Activation | ReLU |
| Optimizer | Adam |

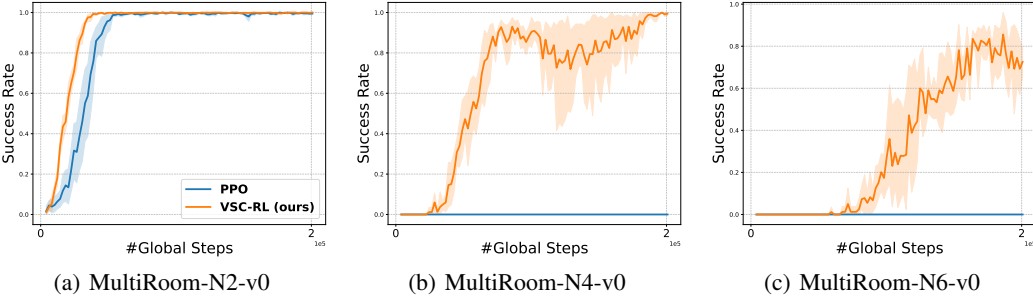

(a) MultiRoom-N2-v0  (b) MultiRoom-N4-v0  (c) MultiRoom-N6-v0

Figure 5: Learning curves on MultiRoom tasks of (a) 2 rooms, (b) 4 rooms, and (c) 6 rooms.

# E  ADDITIONAL EXPERIMENTS: SUBGOAL GENERATOR IN VSC-RL

**Improvement from Subgoal Generator.**  We investigate the importance of the subgoal generator in our VSC-RL on the Web Shopping subsets with different horizon lengths (short, medium and long). We implement VSC-RL with the original goal instead of the subgoals generated from VLM. As shown in Figure 6, the subgoal generator can effectively improve the performance across all types of Web Shopping tasks via autonomously decomposing the original goal into subgoals. Especially, the subgoal generator can effectively enhance the $50\%$ and $32\%$ performance in the Web Shopping medium and long tasks, respectively.

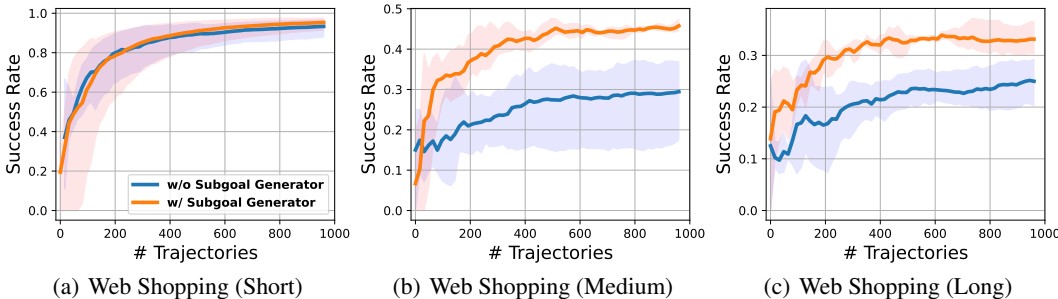

(a) Web Shopping (Short)    (b) Web Shopping (Medium)    (c) Web Shopping (Long)

Figure 6: Success rate on Web Shopping (a) Short, (b) Medium and (c) Long tasks of VSC-RL with and without subgoal generator.

**Verification of Subgoal Generator.**  To investigate the quality and feasibility of the generated subgoals, we manually verify the results of the subgoal generator on 135 trajectories from the AitW human demonstration. There are $135(100\%)$ goals that are decomposed into feasible subgoals successfully, and the final goal can be accomplished by reaching these subgoals sequentially. Specifically, there are $123(91.1\%)$ goals that are decomposed into subgoals completely aligning with the human demonstration. For the remaining $12(8.9\%)$ goals, the subgoal generator provides alternative subgoals different from human demonstration, but still can successfully arrive at the final goal.

# F ADDITIONAL EXPERIMENTS: WEBARENA-LITE

As shown in Table 7, we present the evaluation performance on specific WebArena-Lite tasks, including Reddit, Gitlab, CMS (online store content management system), Map (OpenStreetMap), and OSS (OneStopShop). We also provide the learning curves in Figure 7. These empirical results demonstrate that our VSC-RL achieve the best performance across on all types of tasks, significantly surpassing existing SOTAs.

Table 7: The evaluated performance on WebArena-Lite. The best performance is **highlighted**.

| Method | Task (# Ratio) | | | | | |
|---|---|---|---|---|---|---|
| | Reddit (12.7%) | Gitlab (19.4%) | CMS (21.2%) | Map (18.8%) | OSS (27.9%) | All (100.0%) |
| SFT | 36.8% | 6.7% | 20.0% | 33.3% | 17.8% | 20.6% |
| Filtered BC | 52.6% | 20.0% | 31.4% | 23.3% | 8.9% | 23.0% |
| AWR | 57.9% | 26.7% | 31.4% | 26.7% | 17.8% | 28.5% |
| DigiRL | 52.4% | 28.1% | 37.1% | 32.3% | 15.2% | 30.3% |
| WebRL | 57.1% | 28.1% | 34.3% | **35.5%** | 15.2% | 30.9% |
| VSC-RL (ours) | **61.9%** | **31.3%** | **40.0%** | **35.5%** | **19.6%** | **34.5%** |

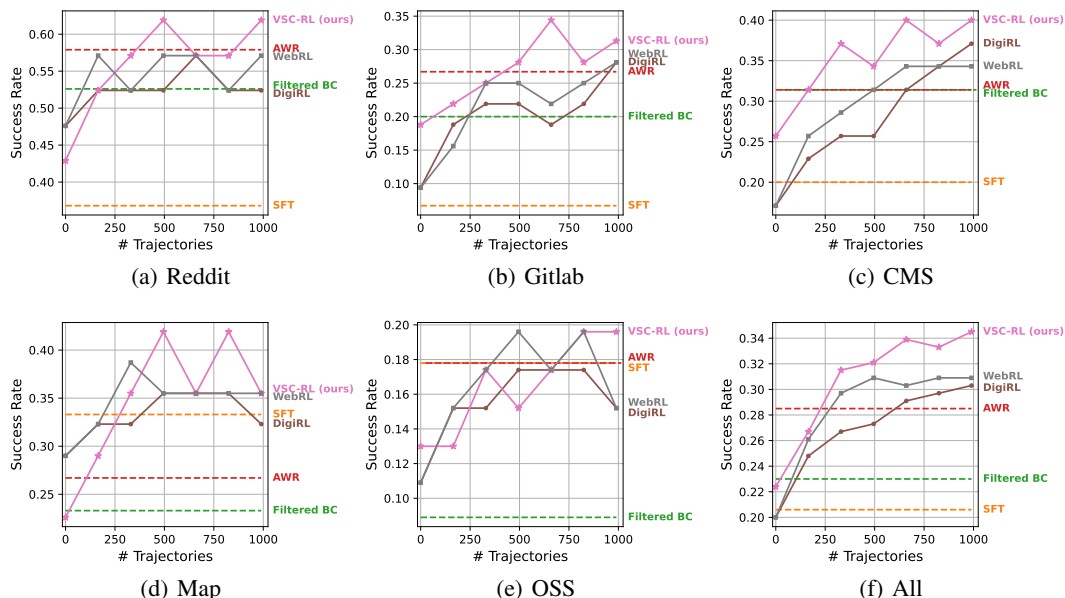

Figure 7: Learning curves on (a) Reddit, (b) Gitlab, (c) CMS, (d) Map, (e) OSS, and (f) All tasks.

## G ADDITIONAL EXPERIMENTS: INCORRECT SUBGOALS

To assess the robustness of VSC-RL with respect to subgoal quality, we conducted an ablation study in which the agent was intentionally provided with incorrect subgoals (e.g., incorrect websites or irrelevant search items) during training. As shown in Table 8, VSC-RL fails to learn effectively under these conditions, resulting in a substantial drop in performance. These results highlight the critical importance of subgoal quality in the learning process and emphasize the necessity of reliable subgoal generation for VSC-RL.

Table 8: Performance of VSC-RL with incorrect subgoals.

|  | Correct | Incorrect |
|---|---|---|
| Train | 64.6% | 19.8% |
| Test | 58.3% | 17.7% |

## H ADDITIONAL EXPERIMENTS: ABLATIONS ON SUBGOAL GENERATOR

We additionally integrated Qwen2.5-VL-72B and Qwen2.5-VL-3B (Bai et al., 2025) into VSC-RL, respectively. As shown in Table 9, Gemini-1.5-Pro consistently outperforms all Qwen2.5-VL models on both the train and test sets. Among the Qwen2.5-VL models, Qwen2.5-VL-72B outperforms Qwen2.5-VL-3B, suggesting it produces higher-quality subgoals. The result indicates that the choice of VLM significantly affects the quality of generated subgoals, which in turn impacts performance. In future work, we plan to fine-tune the open-source VLM for generating subgoals with high quality in the specific domain.

Table 9: Performance comparison with different VLM-based subgoal generators.

|  | Gemini-1.5-Pro | Qwen2.5-VL-72B | Qwen2.5-VL-3B |
|---|---|---|---|
| Train | 64.6% | 60.4% | 58.3% |
| Test | 58.3% | 56.3% | 55.2% |

# I  FAILURE CASES ANALYSIS

We analyze the failure cases of VSC-RL on the Web Shopping test sets. We manually evaluate the failure tasks and summarise the reasons as follows, categorising them as "Stuck Issue", "Wrong Navigation", and "Technical Issue". Specifically, for our VSC-RL,

- "Stuck Issue" (37.5%), common failure modes include being unable to close the Chrome started pop-up (12.5%) and repeatedly typing in the search bar without submitting the query (25.0%). These indicate challenges in interface interaction and accurate action execution.
- "Wrong Navigation" (30.0%), the agent often reaches the correct website but deviates from the target task by entering unrelated categories, showing challenges in interpreting navigation menus.
- "Technical Issue" (32.5%) comprises cases such as getting stuck at CAPTCHA verification (12.5%), page loading failures (2.5%), and LLM misjudgments (7.5%), which reflect either site-level obstacles or model-level perception errors.

To align with the ablation results, Figure 8 shows the failure case distribution in the ablation results of VSC-RL. Removing subgoals and policy gradient mainly increases "Wrong Navigation" errors, while removing imitation loss raises both "Wrong Navigation" and "Stuck Issue".

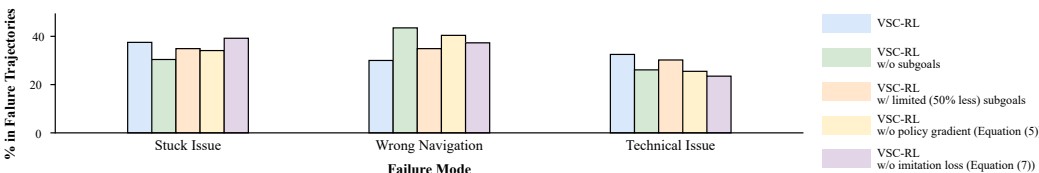

Figure 8: Failure case distribution in the ablation results of VSC-RL on the Web Shopping tasks.

## J PROMPT EXAMPLE

We provide the prompt example of the subgoal generator in our VSC-RL for a given goal and corresponding decomposed subgoals in the MultiRoom (Figure 9), AitW (Figure 10) and WebArena-Lite (Figure 11) tasks.

---

**Prompt**

=====Example=====

**Q:** Based on the provided {*example_image*} of the maze,
   PLEASE provide the sequence of rooms for the agent (RED triangle) to reach
   the goal (GREEN square).

**A:** To reach the goal, the agent need to open to doors to across rooms:
   1. Open the YELLOW DOOR
   2. Open the GREEN DOOR
   3. Open the PURPLE DOOR
   4. Open the YELLOW DOOR
   5. Open the RED DOOR
   6. Reach the GREEN GOAL

*example_image*

=====Your Turn=====

Respond in this format:

**Q:** Based on the provided {*image*} of the maze,
   PLEASE provide the sequence of rooms for the agent (RED triangle) to reach
   the goal (GREEN square).

**A:** To reach the goal, the agent need to open to doors to across rooms
   < DOORS >

**Response**

**Q:** Based on the provided {*test_image*} of the maze,
   PLEASE provide the sequence of rooms for the agent (RED triangle) to reach
   the goal (GREEN square).

**A:** To reach the goal, the agent need to open to doors to across rooms:
   1. Open the BLUE DOOR
   2. Open the GREEN DOOR
   3. Open the YELLOW DOOR
   4. Open the RED DOOR
   5. Open the GREEN DOOR
   6. Reach the GREEN GOAL

*test_image*

---

Figure 9: Prompt example for our subgoal generator for MultiRoom Benchmark. The generator decomposes the goal of navigating the maze into subgoals like opening specific doors sequentially.

**Prompt**

=====**Example**=====

**Q:** Task: Search for "macbook pro" on walmart, select the first entry, and add it to the cart.'

Based on the provided screenshot: {*example_image*},

PLEASE descompse the TASK to the sequence of specific SUBTASKS.

**A:** To complete your requested task, you would need to accomplish these subtasks on a device:

1. Open a browser.

2. Go to the Walmart website.

3. Use the search bar to search for "macbook pro".

4. Click on the first search result.

5. Click on the "Add to Cart" button on the product page.

=====**Your Turn**=====

Respond in this format:

**Q:** Task: {*goal*},

Based on the provided screenshot: {*image*},

PLEASE descompse the TASK to the sequence of specific SUBTASKS.

**A:** To complete your requested task, you would need to accomplish these subtasks on a device:

<SUBTASKS>

*example_image*

**Response**

**Q:** Task: Install the Calendar app

Based on the provided screenshot: {*test_image*},

PLEASE descompse the TASK to the sequence of specific SUBTASKS.

**A:** To complete your requested task, you would need to accomplish these subtasks on a device:

1. Open the Google Play Store app.

2. Search for \"Calendar\" in the Play Store search bar.

3. Select the desired Calendar app from the search results.

4. Click the \"Install\" button.

5. Wait for the app to download and install.

6. Open the app once installed.

*test_image*

Figure 10: Prompt example for our subgoal generator for tasks in AitW dataset. The generator decomposes user commands into actionable subgoals, such as opening a browser, searching for items, and selecting desired results.

**Prompt**

You are an expert web automation agent. Your job is to decompose a user TASK into SUBTASKS for a web agent to follow. Each subtask should be a instruction that represents a meaningful intermediate goal, which can be achieved by performing several low-level actions (not just a single atomic action). Do NOT generate subgoals for steps where the user needs to view, check, or record information. Please reference to the EXAMPLE TASK and EXAMPLE SUBTASKS, and keep the subgoals as close as possible in format, complexity and granularity to the example. If the task involves a date range, please specify the concrete date range in the subgoal like in the example.

=====Website Description=====
Reddit: a web forum and community platform where users can browse, post, and comment on topics in different forums (subreddits). Users can navigate to the Forums page to see a list of all available forums, browse forums by tags or alphabetically, and select a specific forum to view its posts. Within a forum, users can sort posts by criteria such as Hot, New, Top, Controversial, or Most Commented. Each post can be viewed in detail, and users can read or add comments. Users can also submit new posts to a forum. Typical actions include navigating to a forum, sorting and browsing posts, reading and commenting on posts, and submitting new posts.

*example_html*

=====Example=====
**Q:** Task: Among the top 10 post in \"books\" forum, is there any post talks about supporting local book stores? If so, tell me the organizations involved
Based on the provided html: {*example_html*},
PLEASE descompse the TASK to the sequence of specific SUBTASKS.
**A:** To complete your requested task, you would need to accomplish these subtasks on a device:
1. Navigate to the '/f/books' forum page.
2. Sort posts on the '/f/books' forum page by top posts of all time to check the top 10 posts.

=====Your Turn=====
Respond in this format:
**Q:** Task: {*goal*},
Based on the provided html: {*html*},
PLEASE descompse the TASK to the sequence of specific SUBTASKS.
**A:** To complete your requested task, you would need to accomplish these subtasks on a device:
    <SUBTASKS>

**Response**

*test_html*

**Q:** Task: Create a discussion post about \"Iphone 14\" in a relevant subreddit and ask users for their opinions with the simple prompt, \"your opinion\".
Based on the provided html: {*test_html*},
PLEASE descompse the TASK to the sequence of specific SUBTASKS.

**A:** To complete your requested task, you would need to accomplish these subtasks on a device:
1. Navigate to a relevant subreddit for \"iPhone 14\" discussions (e.g., r/apple or r/iPhone).
2. Create a new post in the chosen subreddit with the title \"iPhone 14\" and the body text \"Your opinion\".

Figure 11: Prompt example for our subgoal generator for WebArena-Lite tasks. The subgoal generator decomposes a user task into a sequence of more specific and actionable instructions based on the provided HTML context.

## K  QUALITATIVE EXAMPLE

We provide qualitative examples of VSC-RL applied to MultiRoom (Figure 12), AitW General (Figure 13), AitW Web Shopping (Figure 14), WebArena-Lite Map (Figure 15), and WebArena-Lite CMS (Figure 16).

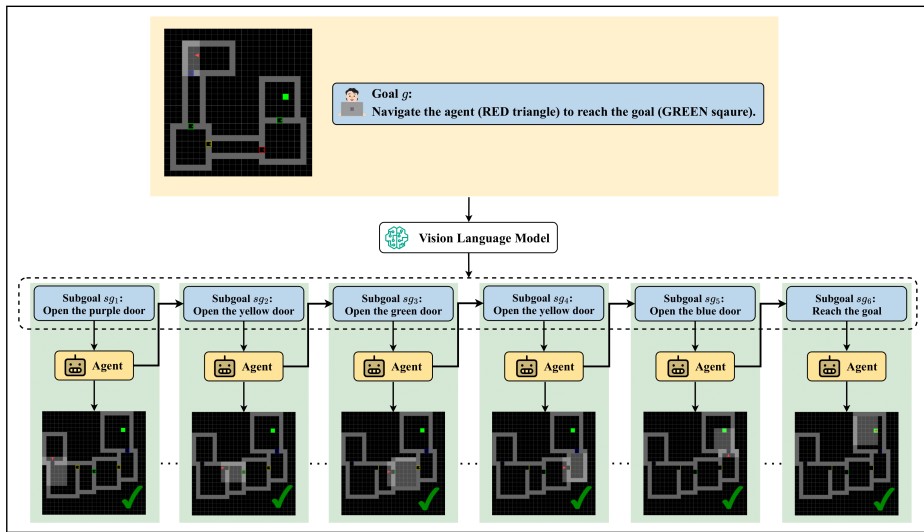

Figure 12: Qualitative example of VSC-RL on the Multiroom task.

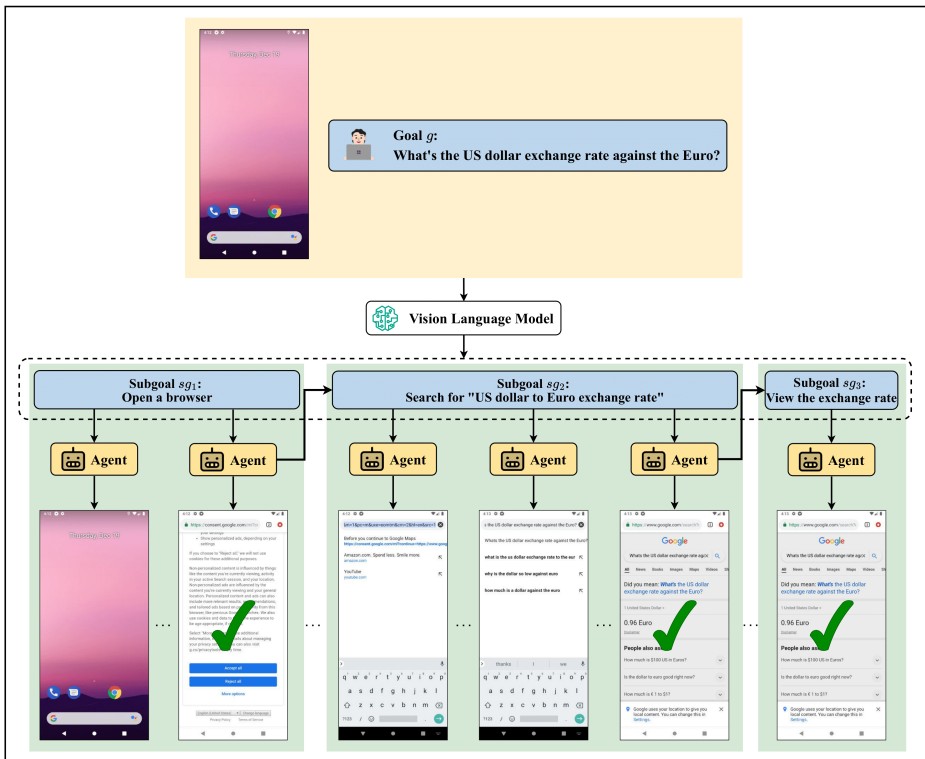

Figure 13: Qualitative example of VSC-RL on the AitW General task.

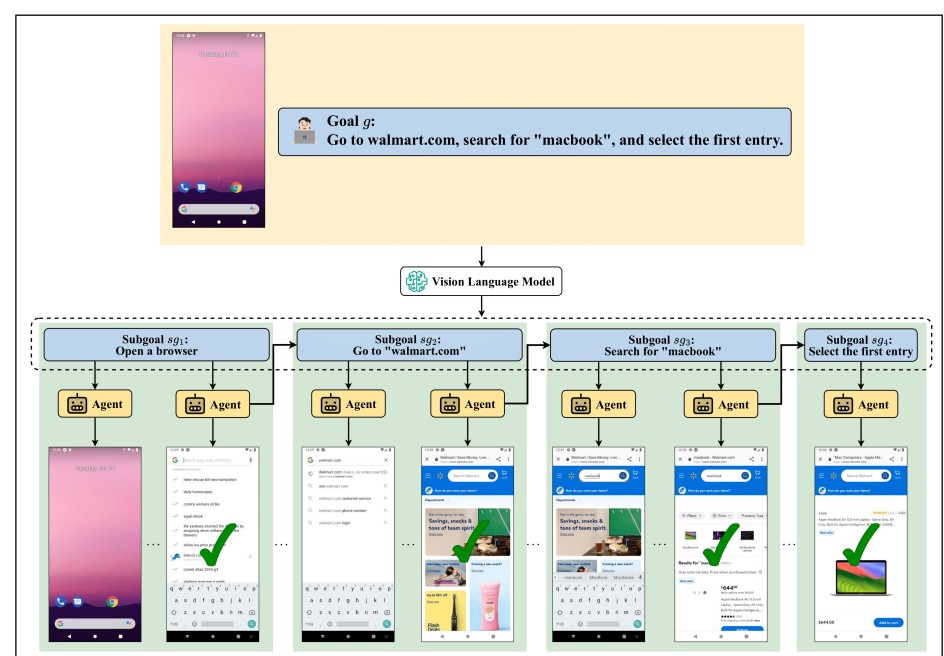

Figure 14: Qualitative example of VSC-RL on the AitW Web Shopping task.

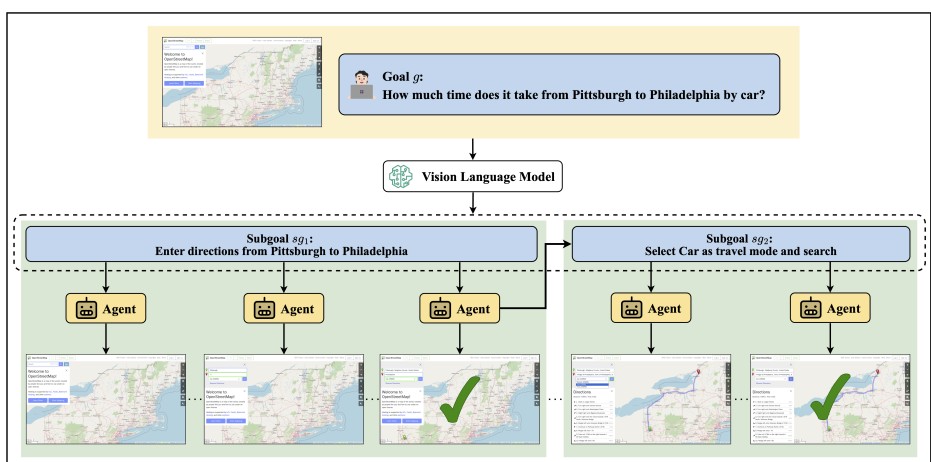

Figure 15: Qualitative example of VSC-RL on the WebArena-Lite Map task.

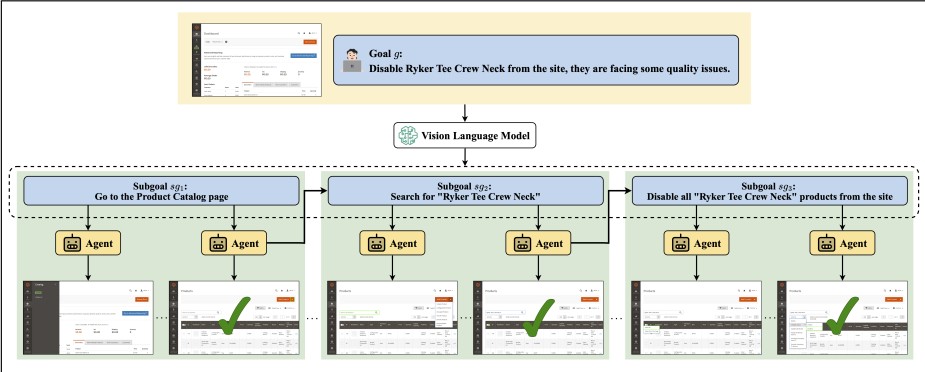

Figure 16: Qualitative example of VSC-RL on the WebArena-Lite CMS task.

