# OpenReview forum: "Advancing Autonomous VLM Agents via Variational Subgoal-Conditioned Reinforcement Learning"
_ICLR.cc/2026/Conference — ICLR 2026 Conference Withdrawn Submission_

### Official Review · Reviewer_znxQ · 2025-10-17

**Soundness:** 2
**Presentation:** 2
**Contribution:** 2
**Rating:** 4
**Confidence:** 3

**Summary:**

VSC-RL pairs a VLM-generated sequence of semantic subgoals with a subgoal-conditioned ELBO that decomposes a trajectory-level KL between a learned policy and a reference policy. The theory claims a reduction from a goal-conditioned ELBO to subgoal terms (Props. 4.1/4.2), yielding the SGC-ELBO (Eq. 3). The algorithm instantiates this with AWR on subgoal-conditioned returns plus an imitation loss on reference actions (Eq. 7). Experiments on AitW (General/Web-Shopping), WebArena-Lite, and MiniGrid show improvements over prompting, BC, and RL baselines.

**Strengths:**

- Conceptual clarity. A clean decomposition from goal-level to subgoal-level terms and timely use of VLMs as high-level scaffold.
- Relevant domains. AitW/WebArena-Lite are appropriate testbeds, and compute reporting is reasonably transparent on the GPU side.
- Some robustness & transparency. There is a corruption test with intentionally wrong subgoals, prompts are provided, and there's an ablation across different VLMs (showing the method is sensitive to subgoal quality).

**Weaknesses:**

1. Eq. (3) and Prop. 4.2 derive a reverse-KL $KL\big(p_{\pi}(\tau\mid\cdot)\big\|p_{\pi_{\text{ref}}}(\tau\mid\cdot)\big)$ at the (sub-)trajectory level, but the implementation uses an imitation loss $\max_{\pi} \mathbb{E}\big[\log \pi(a_{\text{ref}}\mid s, s_g)\big]$ (Eq. 7), which corresponds to a forward-KL surrogate $KL(\pi_{\text{ref}}\|\pi)$. This is not a direct estimator of the theoretically motivated divergence, and the paper neither justifies this approximation nor clarifies the bias it introduces.

2. Experiments use only 3 seeds for AitW and 1 seed for WebArena-Lite, no $\mu \pm$ CI, and error bars are absent from figures. Several baseline results are borrowed from prior papers without matched reruns, making it difficult to assess the true effect sizes and statistical significance of the improvements.

3. Appendix proof errors:
  - In Prop. C.1 "$\log p \propto exp(J/\alpha)$", but if $p \propto exp(J/\alpha)$, then $\log p \propto J/\alpha$ not $exp(J/\alpha)$.
  - In Prop. C.2 "$p_\pi(\tau|g)=\prod_i p_\pi(\tau_i|s_{g_i})$" and then inexplicably write a line "$\le p_\pi(\tau_i|s_{g_i})$".

4. The main environments use 10 steps (General), 20 steps (Web-Shopping), and 30 steps (WebArena-Lite), which are short-to-moderate horizons in the RL literature. The "long-horizon" positioning in the abstract and introduction is not adequately substantiated by these task lengths.

**Questions:**

1. What exact divergence is minimized in practice? Please state explicitly whether Eq. (7) is intended as a forward-KL surrogate and provide a formal justification for the gap to Eq. (3)/Prop. 4.2.
2. Can you run $\geq 5$ seeds (ideally 10) and report mean + confidence intervals?
3. Given that the main tasks use 10/20/30-step horizons, will you either adjust the "long-horizon" framing or add environments with $\geq 50$ steps (or longer subgoal chains) to substantiate this claim?
4. Were subgoals generated online each episode or cached offline? Please release the exact prompts, Gemini model versions, and subgoal transcripts for reproducibility.
5. Can you add oracle upper bounds (perfect subgoals) and subgoal-length sweeps?

---

### Official Review · Reviewer_1sn6 · 2025-10-30

**Soundness:** 2
**Presentation:** 3
**Contribution:** 2
**Rating:** 4
**Confidence:** 4

**Summary:**

The paper proposes VSC-RL to address the low learning efficiency of VLM agents in sparse-reward, long-horizon mobile tasks. VSC-RL improves upon existing RL-based agents by utilizing a VLM (Gemini-1.5-Pro) as a subgoal generator to autonomously decompose complex goals into a sequence of feasible subgoals. It reformulates the decision-making problem as a variational subgoal-conditioned RL problem and thus derives a new optimization objective. Experiments on benchmarks including AitW and WebArena-Lite show that VSC-RL achieves superior learning efficiency and final performance compared to existing methods.

**Strengths:**

1. This paper tackles a significant limitation of current RL-based VLM agents: poor learning efficiency in sparse-reward, long-horizon tasks.
2. This paper provides theoretical derivation of the SGC-ELBO objective that gives principled foundation for the final optimization.
3. Overall, the paper is well-written and clearly organized, making it easy to follow the main ideas and contributions.

**Weaknesses:**

1. The method's core relies on a powerful, proprietary VLM (Gemini-1.5-Pro) to act as the subgoal generator. This introduces significant overhead (requiring Gemini-1.5-Pro calls during both training initialization and subgoal evaluation) and reproducibility issues. The authors acknowledge this limitation but provide no experiments to quantify the performance degradation if a less capable, open-source model were used. Including those results would really help us understand the method's efficiency boundary.
2. This leads to my second question. The agents are all built on AutoUI-Base, which may not be a general-purpose agent and likely lacks task decomposition and planning skills on its own. So, it's not all that surprising that adding a powerful VLM planner on top gives a huge boost. I'm left wondering if VSC-RL would provide the same level of advantage if it were applied to a more powerful, modern base model (like a Qwen2.5-VL) that already has strong planning abilities.
3. The paper compares against baselines from 2023 and 2024. While these are relevant, the field is moving rapidly, and a discussion or comparison with other 2025 era methods for VLM agent training would strengthen the paper's positioning.
4. The paper's primary motivation is solving sparse rewards and long-horizon dependencies. However, the related works section doesn't seem to have a dedicated discussion on the vast body of literature specifically for this challenge (e.g., credit assignment in agent training). It discusses VLM agents and variational RL, but it feels like it's missing a key part of the puzzle.

**Questions:**

See weaknesses

---

### Official Review · Reviewer_txwh · 2025-10-31

**Soundness:** 2
**Presentation:** 2
**Contribution:** 1
**Rating:** 2
**Confidence:** 4

**Summary:**

This work introduces VSC-RL, where it first uses an advanced VLM to composite tasks into subtasks and uses RL to maximize the subgoal-conditioned return. It manages to outperform baseline methods across a set of benchmarks.

**Strengths:**

1. The paper is easy to follow.
2. The paper introduces a novel theoretical framework that reformulates goal-conditioned RL under a variational inference perspective.

**Weaknesses:**

1. The proposed method depends heavily on large, advanced models such as Gemini-1.5-Pro for task decomposition. However, such models already possess strong reasoning and subtask-generation abilities natively. Recent GUI-agent systems (e.g., Operators, Claude CUA) can also perform effective subtask execution without additional hierarchical RL formulations. This raises questions about the necessity and distinct contribution of the proposed method. A more fundamental challenge may instead lie in how to generate appropriate and robust task decompositions automatically.

2. The authors state that Gemini-1.5-Pro is used both for task decomposition and evaluation in AitW . This dual usage introduces a serious concern regarding the reliability and objectivity of the results.

3. Baseline methods appear weak and limited. Several standard methods such as GRPO, PPO, or newer GUI-specific RL frameworks are not reported. The omission of these stronger baselines weakens the empirical validity of the claimed improvements and makes it difficult to assess the true competitiveness of the proposed approach.

**Questions:**

1. Can the authors analyze the challenges involved in training a single model that can both perform task decomposition and generate actions?

---

### Note · Authors · 2025-11-19

**Comment:**

We sincerely appreciate the time and effort that the reviewers have devoted to evaluating our submission. We are grateful for the constructive feedback and thoughtful consideration. After internal discussion, we have decided to withdraw the paper at this time. Thank you again for the opportunity and the valuable comments provided.

**Withdrawal Confirmation:**

I have read and agree with the venue's withdrawal policy on behalf of myself and my co-authors.